# Evaluation of a novel approach to community health care delivery in Ifanadiana District, Madagascar

**Bénédicte Razafinjato[1], Luc Rakotonirina[1], Laura F. Cordier[1], Anna Rasoarivao[1], Mamy Andrianomenjanahary[1], Lanto Marovavy[1], Feno Hanitriniaina[1], Isaïe Jules Andriamiandra[2], Alishya Mayfield[3], Daniel Palazuelos[3,4,5], Giovanna Cowley[1], Andriamanolohaja Ramarson[1], Felana Ihantamalala[1,3], Rado J. L. Rakotonanahary[1,3], Ann C. Miller[1,3], Andres Garchitorena[1,6], Meg G. McCarty[1], Matthew H. Bonds[1,3], Karen E. Finnegan[1,3]***

**1** Pivot, Ranomafana, Fianarantsoa, Madagascar, **2** Madagascar Ministry of Public Health, Antananarivo, Madagascar, **3** Department of Global Health and Social Medicine, Harvard Medical School, Boston, Massachusetts, United States of America, **4** Division of Global Health Equity, Brigham and Women's Hospital, Boston, Massachusetts, United States of America, **5** Partners In Health, Boston, Massachusetts, United States of America, **6** Institut de Recherche pour le Développement, MIVEGEC Laboratory, University of Montpellier, Centre National de la Recherche Scientifique, Antananarivo, Madagascar

* kefinnegan@gmail.com

**Data Availability Statement:** Data availability complies with guidelines as established through the publication and the data sharing agreement

## Abstract

Despite widespread adoption of community health (CH) systems, there are evidence gaps to support global best practice in remote settings where access to health care is limited and community health workers (CHWs) may be the only available providers. The nongovernmental health organization Pivot partnered with the Ministry of Public Health (MoPH) to pilot a new enhanced community health (ECH) model in rural Madagascar, where one CHW provided care at a stationary CH site while additional CHWs provided care via proactive household visits. The program included professionalization of the CHW workforce (i.e., targeted recruitment, extended training, financial compensation) and twice monthly supervision of CHWs. For the first eighteen months of implementation (October 2019-March 2021), we compared utilization and proxy measures of quality of care in the intervention commune (local administrative unit) and five comparison communes with strengthened community health programs under a different model. This allowed for a quasi-experimental study design of the impact of ECH on health outcomes using routinely collected programmatic data. Despite the substantial support provided to other CHWs, the results show statistically significant improvements in nearly every indicator. Sick child visits increased by more than 269.0% in the intervention following ECH implementation. Average per capita monthly under-five visits were 0.25 in the intervention commune and 0.19 in the comparison communes (p<0.01). In the intervention commune, 40.3% of visits were completed at the household via proactive care. CHWs completed all steps of the iCCM protocol in 85.4% of observed visits in the intervention commune (vs 57.7% in the comparison communes, p-value<0.01). This evaluation demonstrates that ECH can improve care access and the quality of service delivery in a rural health district. Further research is needed to assess the

between Pivot and the Madagascar Ministry of Public Health. GIS data from Ifanadiana District are publicly available on OpenStreetMap (https://www.openstreetmap.org); the map base layer can be found at: https://data.humdata.org/dataset/cod-ab-mdg?. Health system data may be shared upon request by contacting science@pivotworks.org.

**Funding:** This work was supported through funding from Pivot to KEF, and from CRI Foundation, MJS Foundation, and Wagner Foundation, and the Herrnstein Family Foundation to Pivot. The funders had no role in study design, data collection and analysis, decision to publish, or preparation of the manuscript.

**Competing interests:** We have read the journal's policy and the authors of this manuscript have the following competing interests: BR, LR, LFC, AR, MA, LM, FH, GC, AR, FI, RJLR, MGM are employees of Pivot. IJA is an employee of the Madagascar Ministry of Public Health. KEF received grant funding from Pivot for this work. MHB is a member of the Pivot board. These interests will not alter adherence to PLOS Global Public Health policies on sharing data and materials.

generalizability of results and the feasibility of national scale-up as the MoPH continues to define the national community health program.

## Introduction

More than half of the world's population lacks access to essential health services [1]. This is especially true for low-income rural communities in Sub-Saharan Africa, where use of primary care health services decreases exponentially with geographic distance [2,3]. The growing movement for universal health coverage (UHC) has been bolstered by a corresponding movement toward strengthened, professionalized community health workers (CHWs) who help address challenges with health care accessibility [4]. However, there is variability in the design, management, and implementation of CHW programs across countries, and often limited evidence on best practice to inform guidelines [5]. There is limited rigorous evidence of best practices for CHW recruitment, length of training and training modalities, supervision, CHW: population ratios, and data collection and use [5–9].

In Madagascar, a country with substantial geographic and financial barriers to care, community health workers (CHWs) provide community-based primary care services [10]. Ministry of Public Health (MoPH) policy requires that there be two CHWs in every fokontany (smallest administrative level comprising of one or several villages, ranging from 400 to 4,500 people) who are elected by their community. CHW tasks include the delivery of integrated community case management (iCCM) for children under-five, malnutrition screening, and community health education. There is no formal education requirement for CHWs, and job-related training varies depending on government and partner support. CHWs often engage in other formal work and are not required to be available to provide healthcare on an established schedule. Under national policy, CHWs do not receive a salary. Instead, they sell medications for a small profit, and thus generate income through a social marketing mechanism. Per national policy, CHWs are supervised through monthly meetings at a health center by the head of the health center if the facility has funds available to support CHW travel for supervision. Effectiveness of Madagascar's CH program has two kinds of challenges: limitations that result from the design of existing policy; and limitations in the implementation of the system based on that policy (i.e. fidelity). Together, the challenges are reflected in inadequate resources, limited supervision, medication stock outs, variable training, and limited data on service provision.

In 2014, the nongovernmental organization (NGO) Pivot began a partnership with the Government of Madagascar to establish a model health district through integrated health system strengthening at all levels of the local public health system in Ifanadiana District. Initial interventions focused on primary health care centers and a district hospital [11]. This has been associated with improvements in the majority of maternal and child health indicators, but with unclear signals on infant and under-five mortality [12]. One explanation for apparent modest effects on mortality is inequity in health care access due to geographic barriers. For example, children living further than 5km from strengthened health facilities accessed care less than once per year even after substantial health system strengthening efforts [3]. Since 2016, Pivot has collaborated with the MoPH to strengthen the community health program with additional training, direct supervision, modest compensation, and support with infrastructure, equipment, and supplies. However, challenges related to recruitment, patient access, and supervision were identified. As a result, in 2019, the MoPH and Pivot initiated a redesigned

enhanced community health (ECH) program guided by global best practices on CHW optimization regarding recruitment, compensation, supervision, and training [13]. This program included proactive case-finding through monthly household visits combined with ongoing provision of care at fixed community health sites. We evaluate the impact of this ECH model compared to the existing strengthened program (through Pivot support) using routinely collected program data during the first year of implementation.

## Methods

Ifanadiana is a rural district located in the southeast of Madagascar with a population of 182,000, nearly 33,000 of whom are children under five. The district is composed of 15 communes and 195 fokontany. In 2014, the baseline of a population-representative household survey found under-five mortality was 145 per 1,000 live-births [14]. The follow-up 2018 study found prevalence of 9.1% of diarrhea, 20.1% of fever, and 16.1% of cough and difficulty breathing [in the two weeks preceding the survey) among children under-five in the district [15] (S1 Table).

In 2016, Pivot began supporting the community health program in select communes of Ifanadiana District. In 2019, guided by the principles put forth by the World Health Organization and the CHW Assessment and Improvement Matrix (AIM) Framework [5,13], Pivot proposed an ECH pilot in the Ranomafana commune of Ifanadiana District, where CHWs had been underperforming relative to other communes (Table 1). Ranomafana commune consists of eight fokontany, with an estimated population of 11,960, including 2,150 children. ECH is part of a strategy to achieve UHC through expanded access to high quality services [13]. In partnership with the MoPH, ECH focused on 1) professionalization of CHWs; 2) improved care delivery through a two-pronged approach; and 3) reinforcement of the health management information system (HMIS).

**Table 1. Summary of community health intervention for intervention area, control area, and the rest of the country under Madagascar's national program with key differences of ECH bolded.**

|  | Madagascar national program | Comparison communes: Pivot support of national program | Intervention commune: Enhanced community health pilot |
|---|---|---|---|
| Staffing | • 2 CHWs per fokontany | • 2 CHWs per fokontany | **3–5 CHWs per fokontany** depending on geographic spread and population |
| CHW recruitment strategy and selection criteria | • Elected by community<br>• Literate<br>• Predominantly male | • Elected by community<br>• Literate<br>• Predominantly male | • Jointly recruited by NGO and community<br>• Literate<br>**Gender parity** |
| Training | • Inconsistently trained in iCCM and family planning, depending on partner and MoPH support | • Consistently trained in iCCM and family planning, community malnutrition, tuberculosis screening | **Trained in new protocols for proactive doorstep care,** includes iCCM and family planning, community malnutrition, tuberculosis screening |
| Supervision | • Monthly group supervision at health center, depending on resources | • Monthly group supervision at health center<br>• Quarterly community-based supervision | Monthly group supervision at health center<br>**Monthly community-based supervision** |
| Compensation | • Per diems for training and supervision<br>• Profit through sale of medication | • Per diems for training and supervision<br>• Stipend equal to profit for medication sale (CHWs do not sell medications) | • **Salary** equivalent to Madagascar's minimum wage |
| Workflow | • Work from fixed location (often their home)<br>• Availability variable | • Work from fixed community health posts<br>• Availability variable | **Visit every household once per month**<br>• Community health post remains open<br>**CHWs work full-time** |
| Data collection | • Varies | • Complete iCCM form, register, and monthly activity report | Complete iCCM form, **adapted register that record household visits,** and monthly activity report |
| User Fees | • Free consultation, fee for medicine | • No fees to patients | • No fees to patients |

As Table 1 indicates, key components of the strategy to professionalize CHWs included targeted recruitment, compensation, increased supervision, and additional training. The number of CHWs in each fokontany were increased based on population density. Newly recruited CHWs were active in the community, full-time residents, physically capable of traveling from house to house, and had a gender balance with the goal of achieving parity across the CHW cohort. The total number of CHWs across the 8 fokontany of the intervention commune increased from 16 to 28 (3–5 per fokontany), an average of one CHW per 427 people. CHWs received a monthly salary equivalent to Madagascar's minimum wage and were formally evaluated every 6 months (delayed in 2020 due to COVID-19), which included a review of productivity and quality of care. In this pilot program, CHWs could be terminated if they were not performing adequately, in contrast to the national program where termination of elected CHW was determined by local political leaders.

Pivot created a new cadre of community health supervisors to support the implementation of the ECH program. Community health supervisors are health care workers with a degree in nursing or midwifery, and with experience in community health. Supervision included two activities per month: 1) community-based supervision in which the CHW and supervisor met one-on-one, where the supervisor provided feedback on observed sick child visits; and 2) health center-based group supervision which included discussion of activities, review of data, and training on new tools or methods. During community-based visits, supervisors also provided community health education and care for sick children >5 years old, whose care was not part of the iCCM protocol or national policy.

Improved care delivery included a redesign of service delivery and workflow. The two-pronged approached in ECH commune required CHWs to provide care both at the CH site and through proactive household visits. One CHW in each fokontany was stationed at the CH site while other CHWs traveled to a circuit of homes. The redesign of care delivery was intended to overcome geographic barriers (e.g. distance, time, cost) which prevent patients from seeking care from CHWs, and to formalize the relationship between CHWs and the households, while also maintaining a functional CH site [16–18]. During household vsits, the CHWs actively sought out sick children and followed up on children previously diagnosed with malaria, diarrhea, pneumonia, or malnutrition. CHWs followed up with sick children three days after diagnosis of any illness to determine if symptoms had resolved and, for severe cases where referral was required, CHWs visited the next day to ensure that the sick child had visited the health center.

To strengthen the CH HMIS system, program managers oversaw development and use of new data collection and management tools. During field-based visits, supervisors reviewed data collection processes and completion of forms to ensure that CHWs were collecting high quality data on their activities. At monthly health-center based supervisions, CHWs reviewed and submitted monthly aggregate activity reports.

The impact of the program was evaluated using the data that were routinely collected as part of the program. Data were extracted on the number of children seen at CH sites and via proactive home visits from monthly CHW activity reports to measure CHW utilization by children under-five; a visit was defined as a new or follow-up clinical encounter for sick child care. Information on adherence to the iCCM protocol, a proxy measure for quality of care, came from this same monthly report. The monthly report includes measures of iCCM protocol adherence for all CHW visits: correct treatment of diarrhea (child with diagnosis of simple diarrhea provided with ORS), respiratory infection (child with diagnosis of suspected pneumonia treated with amoxicillin), malaria (child with malaria diagnosed by rapid diagnostic test treated with ACT), and speed of fever treatment (child with fever seen by CHW within 24 hours of fever onset). There was also data obtained on CHW adherence to the iCCM protocol

from the observation checklist completed by CHW supervisors during field-based direct supervision visits. Using data from the observation checklist, we calculated a summary measure of quality of care, correct care, which is defined as how many of the total steps of the iCCM protocol the CHW completed correctly based on the child's diagnosis; this includes assessment, disease classification, treatment, and counseling of the caregiver. Correct care was only calculated following the initiation of ECH as a revised supervision tool was introduced as part of the CH program redesign.

A quasi-experimental study design was used to determine the impact of ECH. We compared outcomes in the intervention area before and after the implementation of ECH with outcomes from five other communes in Ifanadiana. These other communes were supported by Pivot and the MoPH as defined in Table 1 as part of a broader health system strengthening partnership. A summary of health system readiness and disease burden in the intervention and comparison communes can be found in Supplemental Information. It was not possible to compare the intervention area to the national model because data from CHWs registries were not reliably maintained and reported.

Program cost data were extracted from the financial program records of the NGO which oversaw implementation. All financial expenditure data were recorded in QuickBooks.

The ECH pilot began in October 2019. Utilization and program delivery were assessed in the intervention and comparison communes before and after the start of the intervention. A two-sample t-test was used to compare continuous outcomes and a chi-squared test for categorical outcomes. We estimated the difference-in-differences in monthly consultations with children under-five using a linear regression. We compared correct care using a t-test. Data analysis was completed in R version 3.5.2.

This study was approved by the Secretary General of the Ministry of Public Health of Madagascar and was determined to be not human subjects research by Harvard Medical School's Institutional Review Board because it is based on aggregated, routinely collected data.

## Results

Although the ECH intervention commune had low performance prior to the intervention, after the intervention, it significantly outperformed the strengthened comparison communes in nearly every CHW performance metric. During the eighteen-month study period, CHWs in the intervention commune completed 9,652 visits with children under-five, representing a 269.0% increase in consultations from the eighteen months prior. Mean monthly consultations for children under-five increased significantly more for the ECH intervention area than for the comparison area (DID estimate 149.85 [95% confidence interval 49.3, 250.4], p-value <0.01) (S1 Text). Average monthly per-capita utilization in the ECH area was 0.25, which corresponds to 2.9 visits per capita per year (Fig 1). Average monthly per capita utilization was 0.19 in the comparison group for the same eighteen-month intervention period. Per capita utilization increased over time across all eight fokontany in the intervention commune (Fig 2).

In the intervention commune, 40.3% of CHW consultations were proactive at the household (Fig 3); on average, 85.9% of households were visited at least once every month. CHWs in the intervention commune evaluated 6,762 cases of fever (70.1% of visits), 1,153 cases of diarrhea (11.9% of visits), 4,789 cases of pneumonia (49.6% of visits), and 1,407 cases of cough or cold (14.6%) during the eighteen-month period (some children were diagnosed with more than one illness during their visit). In the comparison communes, CHWs completed a total of 33,942 consultations with children under-five during the study period. Of these visits, 83.1% were for fever, 10.1% for diarrhea, 47.0% with pneumonia, and 11.7% with cough or cold.

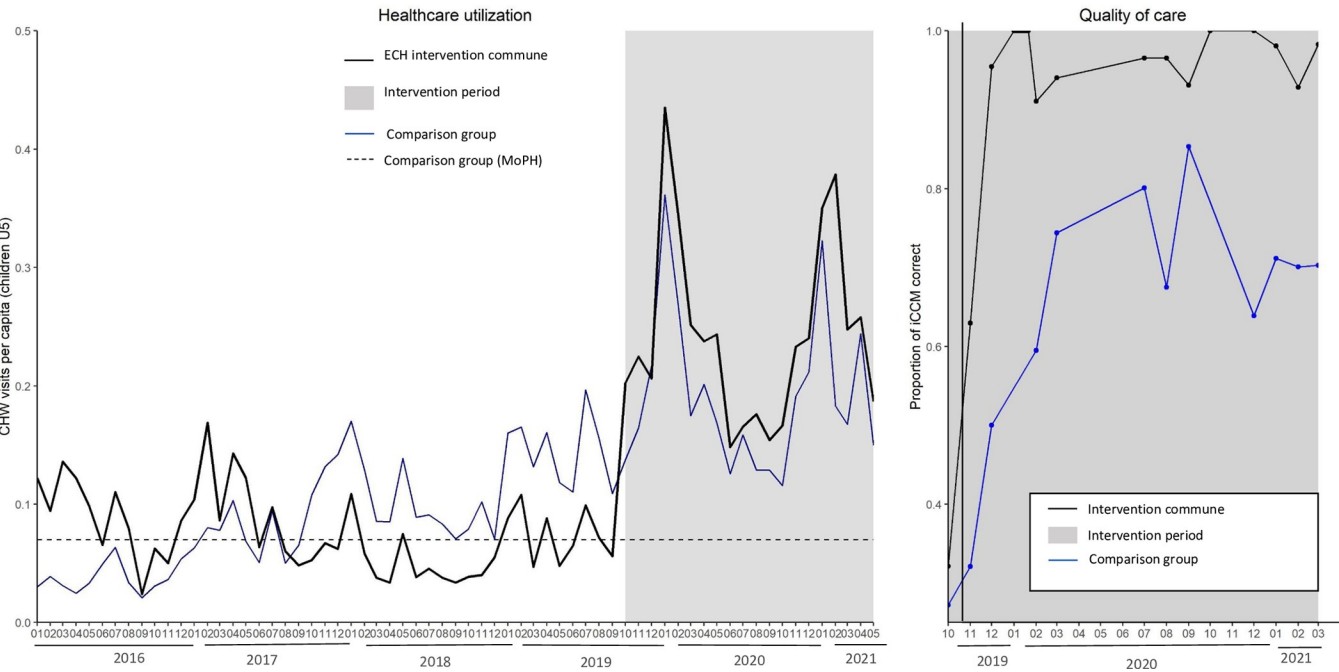

**Fig 1.** Average monthly per capita utilization by children under-five of CHWs by intervention group from January 2016 to March 2020 (left). A comparison of correct care (right), measured through direct observation of the CHW during an iCCM, in the intervention commune (black) and comparison communes receiving enhanced standard of care (blue) from October 2019 to March 2020.

Correct care, as measured through supervisor observation, increased from 32.0% in October 2019 to 98.3% in March 2021 in the intervention commune. Over the eighteen-month study period, in 85.4% of observed visits, CHWs provided care which was consistent with all aspects of the iCCM protocol in the intervention commune; in the comparison communes, CHWs demonstrated complete protocol adherence in 57.7% of observed visits (p-value of difference <0.01) (Table 2). Likewise, CHWs in the intervention commune demonstrated better evaluation of danger signs (97.6% vs 91.2% in comparison area, p-value<0.01), correct treatment of illness (98.1% vs 92.6%, p-value<0.01), and counseling of caregiver on treatment and disease management (89.4% vs 73.2%, p-value<0.01). CHWs in intervention and comparison communes demonstrated similarly high rates of correct diagnosis of illness. All measures of quality improved sharply in the intervention area in the first three months of the study period and remained high. By the end of the first six months of implementing enhanced care, the quality of iCCM care provided by newly recruited CHWs was equal to that of existing CHWs who had already been working in community health (S2 Table).

Disease-specific quality of care measures varied over time (Fig 4). During the eighteen-month study period, the intervention commune provided higher rates of treatment of malaria (intervention = 79.3%, comparison = 57.1%, p-value = 0.04), pneumonia (intervention = 65.0%, comparison = 56.3%, p-value = 0.14), and diarrhea (intervention = 88.3%, comparison = 69.3%, p-value<0.001). Only 10.3% of fever cases were seen within 24 hours of symptom onset in the intervention commune, compared to 12.9% in the comparison communes during the study period.

Supervisors completed 376 supervision visits in the intervention area and directly observed 636 sick child visits to document quality of care and provide feedback. Nearly all (94.7%) of CHWs were supervised twice per month for 14 of 18 months during the study period; supervision was suspended April-June 2020 due to concerns about COVID-19 and in November

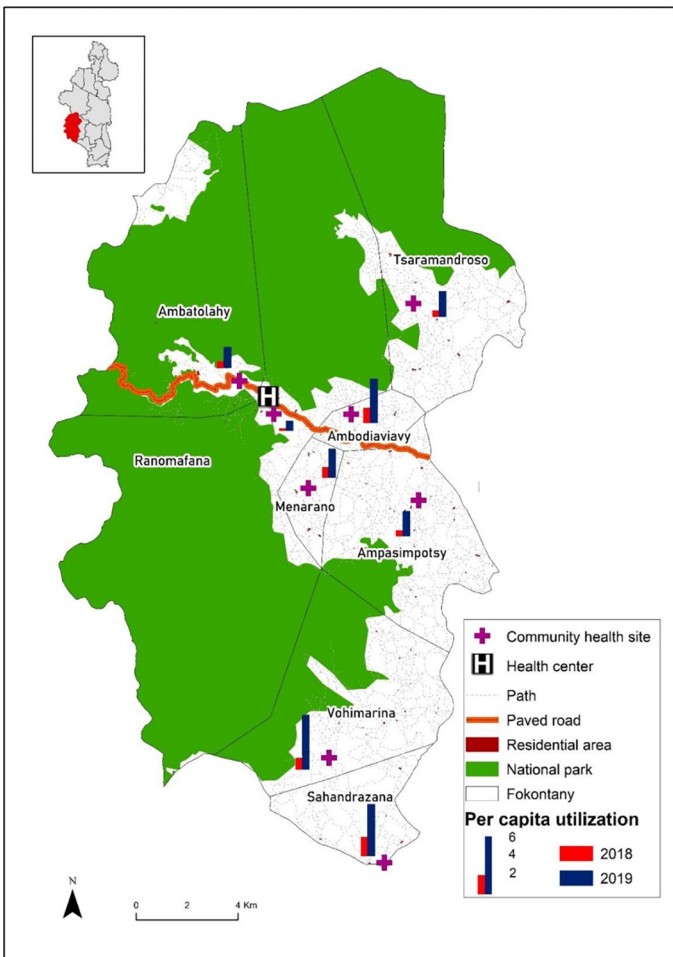

**Fig 2. Per capita community health utilization by fokontany in the intervention commune in October 2018-March 2019 (red) and October 2019-March 2020 (blue).** The district of Ifanadiana is in the inset with the intervention commune in red. The base map layer comes from https://data.humdata.org/dataset/cod-ab-mdg.

2020. In the comparison communes, supervisors completed 1,176 community supervision visits. On average, 66.5% of CHWs in comparison communes received community-based supervision per month.

## Data quality

In both intervention and comparison communes, the monthly CHW activity report was submitted on time and without missing data. Over the study period, the concordance rate between the iCCM form (used to record detailed data during the patient visit), the iCCM register (a line listing summary of each patient visit), and the monthly report (an aggregate summary of the activity of all CHWs in a commune) was 90.8% in the intervention commune and 89.5% in comparison areas. Higher rates of concordance indicate data which is consistent across sources.

## Program costs

The estimated cost of the ECH model is $4.45 per capita for the first year of implementation, compared to $2.32 per capita for the care model in the comparison communes. ECH model

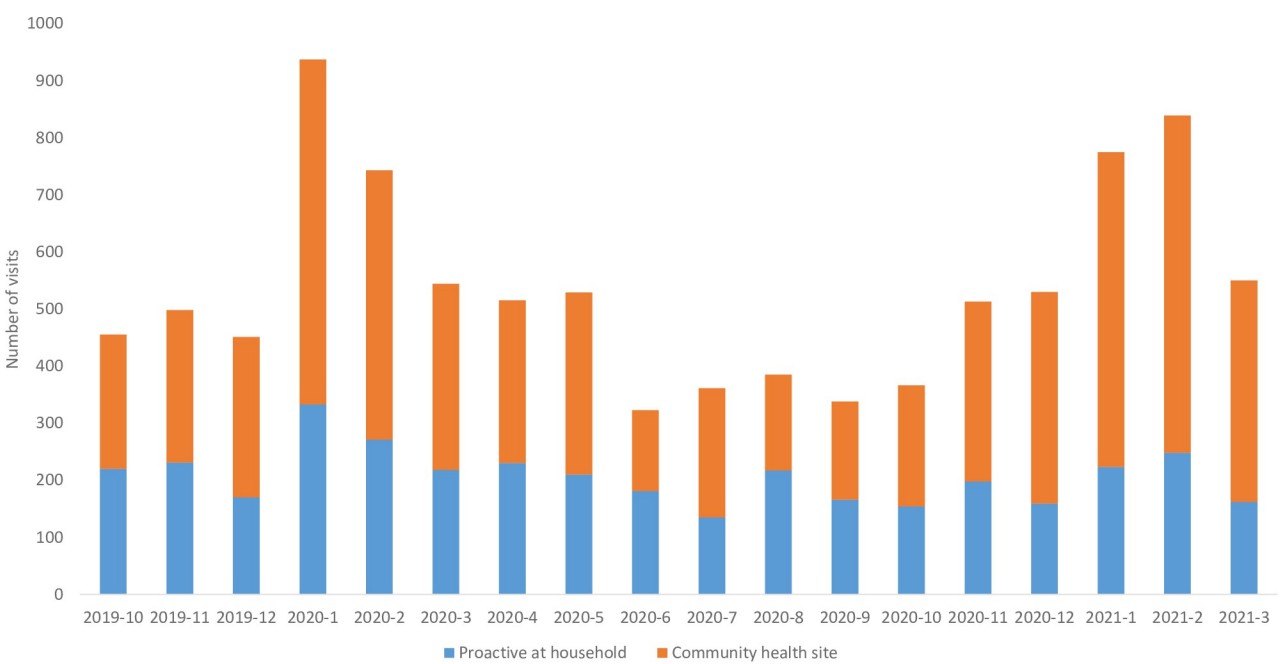

**Fig 3. CHW visits by location in the intervention commune during the implementation of ECH.** Household visits are in (blue) and CH site visits are in (orange).

costs include startup costs of training and equipping new CHWs (e.g. medical supplies, equipment for traveling on foot through hilly, wet terrain) and salary support. This cost excludes mobile technology development and implementation, which took place in the second year of the project at the end of this study period.

## Discussion

Here, we evaluated the performance of an enhanced community health program. Due to lack of data availability, this performance is not compared to the national standard of care, which is substantially under-resourced and supported. Instead, it is compared to communes with strengthened CHWs that, while locally elected and operating from a designated location in accordance with national policies, receive technical and financial support from an NGO for supervision and training, as well as modest compensation which ensures consultations are free of charge. The ECH model includes both proactive home visits and health post-based care by professionalized CHWs and was implemented in rural Madagascar. The program

**Table 2. Quality of care of iCCM provided by CHWs during observed visits from October 2019-March 2020.**

|  | Intervention commune | Comparison communes | p-value |
| --- | --- | --- | --- |
| Number of clinical encounters observed | 636 | 2166 |  |
| Children correctly cared for according to iCCM protocol | 85.4% | 57.7% | <0.01 |
| Correct evaluation of danger signs | 97.6% | 91.2% | <0.01 |
| Correct diagnosis of illness | 99.4% | 97.9% | 0.10 |
| Correct treatment of illness | 98.1% | 92.6% | <0.01 |
| Caregiver counseled | 89.4% | 73.2% | <0.01 |
| Child correctly referred to health center | 94.1% | 80.8% | 0.01 |

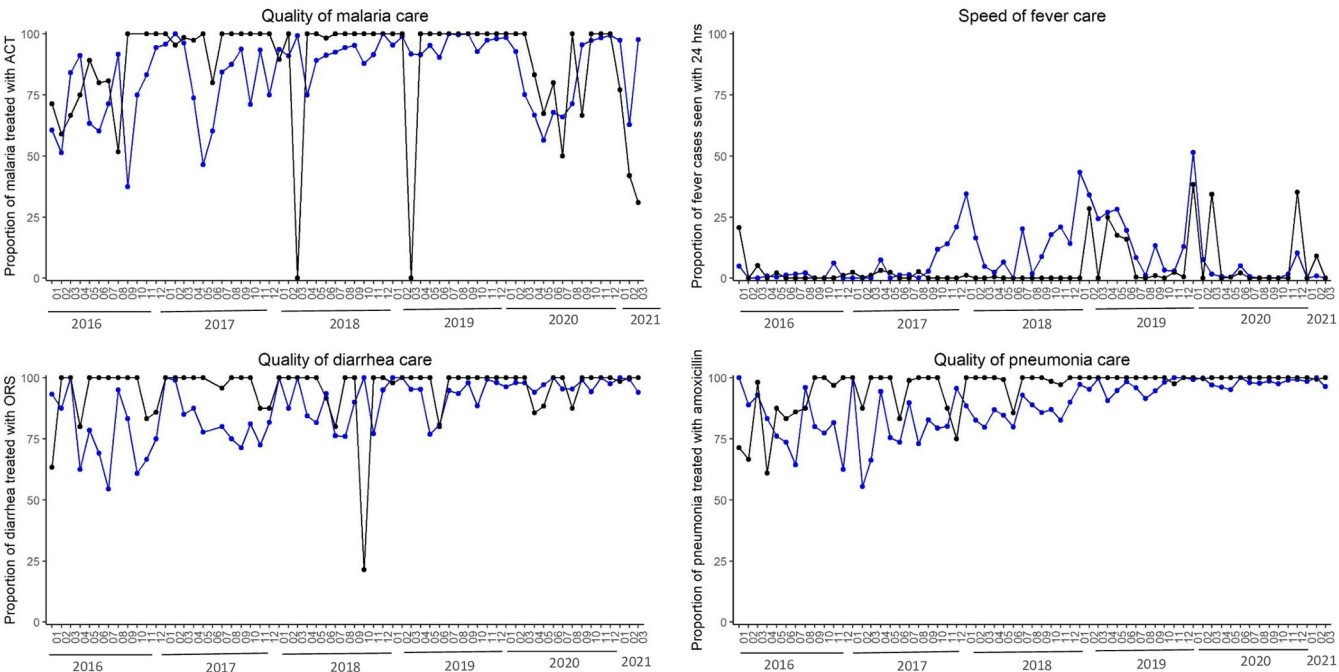

**Fig 4.** Disease-specific quality of care measures from iCCM patient visits for malaria treatment with ACT (top left), speed of fever treatment (top right), diarrhea treatment with ORS (bottom left), and pneumonia treatment with amoxicillin (bottom right) comparing the intervention commune (black) with the average of the quality measure from the five comparison communes (blue).

demonstrated improvements in every measure of service delivery and quality of care over time and in comparison to other communes that had more modestly strengthened community health systems. Evaluation of the program provides lessons for national stakeholders and program managers on the impact of program re-redesign and reveals important considerations for scale-up.

Over the eighteen-month study period, almost half of visits completed by CHWs in the intervention commune were proactive household visits. The increase in per capita utilization during the intervention period provides clear evidence that the two-pronged approach to care increased access. This is further supported by the increase in per capita utilization in fokontany which are farthest from the health center. Both areas of intervention had high rates of referral, but the data do not indicate if the referrals were completed. This information is important as CHWs refer patients for higher levels of care when danger signs are present or when they are unable to treat the illness.

The enhanced community health program introduced fully professionalized CHWs to the community health system. Initially, the program required intensive program management and placed high performance expectations on CHWs and supervisors. To ensure that CHWs were supervised twice per month, supervisors from other communes were called to the intervention area to provide community-based supervision. Although this was effective in meeting supervision targets and improving quality of care, sustaining such intensive intervention is a key challenge; indeed, supervision rates were challenging to maintain during the COVID-19 surge and under other program pressures. Supervision often requires supervisors to travel for multiple days on foot to reach CHWs in remote communities. Under national guidelines, supervision of CHWs is the responsibility of health center managers. The development of a cadre of CHW supervisors is one innovation of the ECH model that helped address the time limitations of

health center staff who are needed for direct provision of clinical care. The World Health Organization names supportive supervision as an important component of CH programs and research highlights the importance of supervision in establishing effective high quality care, although evidence on the impact of supervision on quality of care is mixed [7,19]. Although the development of a cadre of dedicated supervisors was important for the launch of ECH, for long term implementation and scale up, Pivot has proposed training high-performing CHWs to serve as peer mentors and perform many of the functions of supervisors.

Community health workers included in the ECH intervention demonstrated improvements in utilization and quality of care. It is not possible to isolate effects of any single element of the program on outcomes. It is possible that these changes were influenced by increasing familiarity with protocols over time and not program changes. Additionally, as supervision included support on record-keeping and data submission, it is possible that changes over time are a reflection of better reporting by CHWs. Supervision included support for reporting in both ECH and comparison communes.

The estimated costs of ECH model–roughly double that of the comparison commnunes— are consistent with those found in other settings [20]. When studying Madagascar's national community health program, Brunie et al found that CHWs reported high levels of satisfaction with their work, but also high levels of financial uncertainty and most relied on subsistence farming for their livelihoods [21]. Under the ECH model in Ranomafana, CHWs received a financial incentive equivalent to Madagascar's minimum wage (approximately $70 USD) per month, which helped alleviate these financial concerns. Sustainability and harmonization are key factors when considering CHW salary as part of iCCM programs in low resource countries [22]. This evaluation provides evidence of the feasibility of providing remuneration as part of program management although more research is necessary for national scale-up.

There are pressing questions about how CHW programs should be designed and implemented. Relevant global evidence is mixed and often weak [19]. Using data collected as part of MoPH reporting and program documentation, our evaluation of enhanced community health in a rural commune in Madagascar can contribute to global evidence on the optimal design of community programs. Moreover, this experience can provide actionable lessons for Madagascar's national community health program on program design to align with global best practice on community health program principles and contribute to the country's objectives around universal health coverage.

## Supporting information

**S1 Table. Health system readiness and disease burden in intervention and comparison communes.**
(XLSX)

**S2 Table. Comparison of newly hired and existing CHWs.**
(XLSX)

**S1 Text. Interrupted time series analysis.**
(DOCX)

## Acknowledgments

We would like to thank the health system professionals who helped with the conception, implementation and realization of community health activities: Dr Jafeta Andriantahina (Director of Basic Health Care), the health center staff in Ranomafana, the district health team,

the Ministry of Health, the supervisors, and especially the CHWs, supervisors, and communities of Ifanadiana District.

## Author Contributions

**Conceptualization:** Luc Rakotonirina, Laura F. Cordier, Alishya Mayfield, Daniel Palazuelos, Giovanna Cowley, Ann C. Miller, Andres Garchitorena, Meg G. McCarty, Matthew H. Bonds, Karen E. Finnegan.

**Data curation:** Bénédicte Razafinjato, Anna Rasoarivao, Mamy Andrianomenjanahary, Andriamanolohaja Ramarson, Rado J. L. Rakotonanahary, Karen E. Finnegan.

**Formal analysis:** Bénédicte Razafinjato, Karen E. Finnegan.

**Funding acquisition:** Karen E. Finnegan.

**Investigation:** Bénédicte Razafinjato, Laura F. Cordier.

**Methodology:** Laura F. Cordier, Meg G. McCarty, Matthew H. Bonds.

**Project administration:** Bénédicte Razafinjato, Lanto Marovavy.

**Resources:** Mamy Andrianomenjanahary, Feno Hanitriniaina, Isaïe Jules Andriamiandra, Daniel Palazuelos, Giovanna Cowley, Andriamanolohaja Ramarson, Meg G. McCarty.

**Validation:** Bénédicte Razafinjato.

**Visualization:** Bénédicte Razafinjato, Laura F. Cordier, Andres Garchitorena, Matthew H. Bonds, Karen E. Finnegan.

**Writing – original draft:** Bénédicte Razafinjato, Luc Rakotonirina, Laura F. Cordier, Alishya Mayfield, Meg G. McCarty, Matthew H. Bonds, Karen E. Finnegan.

**Writing – review & editing:** Bénédicte Razafinjato, Luc Rakotonirina, Laura F. Cordier, Anna Rasoarivao, Mamy Andrianomenjanahary, Lanto Marovavy, Feno Hanitriniaina, Isaïe Jules Andriamiandra, Alishya Mayfield, Daniel Palazuelos, Giovanna Cowley, Andriamanolohaja Ramarson, Felana Ihantamalala, Rado J. L. Rakotonanahary, Ann C. Miller, Andres Garchitorena, Meg G. McCarty, Matthew H. Bonds, Karen E. Finnegan.

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
