## [Decision Letter · Decision Letter 0]

13 Apr 2023

PGPH-D-22-01753

Evaluation of a novel approach to community health care delivery in Ifanadiana District, Madagascar

Dear Dr. Finnegan,

Thank you for submitting your manuscript to PLOS Global Public Health. After careful consideration, we feel that it has merit but does not fully meet PLOS Global Public Health’s publication criteria as it currently stands. Therefore, we invite you to submit a revised version of the manuscript that addresses the points raised during the review process.

We look forward to receiving your revised manuscript.

Kind regards,

Abraham D. Flaxman, Ph.D.

Academic Editor

Journal Requirements:

1. Please amend your detailed online Financial Disclosure statement. This is published with the article. It must therefore be completed in full sentences and contain the exact wording you wish to be published.

a) Please clarify all sources of funding (financial or material support) for your study. List the grants (with grant number) or organizations (with url) that supported your study, including funding received from your institution. 

b) State the initials, alongside each funding source, of each author to receive each grant.

c) State what role the funders took in the study. If the funders had no role in your study, please state: “The funders had no role in study design, data collection and analysis, decision to publish, or preparation of the manuscript.”

d) If any authors received a salary from any of your funders, please state which authors and which funders.

2. Please declare all competing interests beginning with the statement "I have read the journal's policy and the authors of this manuscript have the following competing interests:"

4. Some material included in your submission may be copyrighted. According to PLOS’s copyright policy, authors who use figures or other material (e.g., graphics, clipart, maps) from another author or copyright holder must demonstrate or obtain permission to publish this material under the Creative Commons Attribution 4.0 International (CC BY 4.0) License used by PLOS journals. Please closely review the details of PLOS’s copyright requirements here: PLOS Licenses and Copyright. If you need to request permissions from a copyright holder, you may use PLOS's Copyright Content Permission form.

Potential Copyright Issues:

Figure 2: please (a) provide a direct link to the base layer of the map (i.e., the country or region border shape) and ensure this is also included in the figure legend; and (b) provide a link to the terms of use / license information for the base layer image or shapefile. We cannot publish proprietary or copyrighted maps (e.g. Google Maps, Mapquest) and the terms of use for your map base layer must be compatible with our CC-BY 4.0 license. 

Additional Editor Comments (if provided):

Here are my thoughts on a few additional changes that I think would enhance the impact of this work for future readers:

In Table 1, is there a way to highlight the key differences between the columns for each row?

The costing evidence is valuable enough that I would propose you move it from the discussion to the results section (and potentially add some description of the approach to gathering this data to the methods section). You could potentially also present it in an additional table in the results section, to make sure busy readers notice it. Of course, your discussion of the implication of these costs is still appropriate to include in your discussion!

Reviewers' comments:

Reviewer's Responses to Questions

**Comments to the Author**

1. Does this manuscript meet PLOS Global Public Health’s publication criteria? Is the manuscript technically sound, and do the data support the conclusions? The manuscript must describe methodologically and ethically rigorous research with conclusions that are appropriately drawn based on the data presented.

Reviewer #1: Yes

2. Has the statistical analysis been performed appropriately and rigorously?

Reviewer #1: Yes

3. Have the authors made all data underlying the findings in their manuscript fully available (please refer to the Data Availability Statement at the start of the manuscript PDF file)?

Reviewer #1: Yes

4. Is the manuscript presented in an intelligible fashion and written in standard English?

Reviewer #1: Yes

5. Review Comments to the Author

Reviewer #1: Overall, the study provides a clear summary of a quasi-experimental study that evaluated the impact of a new enhanced community health (ECH) model on the utilization and quality of integrated community case management in a rural health district in Madagascar. The use of specific indicators, such as sick child visits and per capita monthly under-five visits, helps to demonstrate the effectiveness of the ECH model.

One limitation of the study is that the quasi-experimental design cannot establish causality, and other factors besides the ECH model may have contributed to the observed improvements in utilization and quality of care. I think the study can be strengthened by additional examination such as performing an interrupted time series analysis with control to rule out alternative explanations for observed changes in the dependent variable over time instead of only conducting t-test and chi-squared test.

The study acknowledges the need for further research to assess the generalizability of the results and the feasibility of national scale-up, which is a critical step in the implementation of effective community health programs. In general, the study provides a useful summary of a study that provides evidence for the potential benefits of the ECH model in improving access to care and quality of service delivery in remote settings where access to health care is limited.

Other suggestions or questions:

1. The findings can be enhanced by quantifying the effects of intervention through a difference-in-difference analysis.

2. It would be helpful to conduct an equivalent regression analysis to produce uncertainty estimates on the results. E.g., line 34.

3. Figure 1 shows that the health utilization of both intervention and comparison group has increased after the intervention was implemented, is there any explanation for that? Using an interrupted time series design with a control group could increase the validity of the study results.

4. It would be helpful to discuss other possible factors that might have contributed to the performance of the intervention group to increase.

5. Is it possible that the recruitment of more skillful CHWs from the comparison commune to the intervention commune, due to higher compensation, have caused the differences in outcomes between the two groups?

6. Discussing and justifying why the control groups are comparable to the intervention group would be helpful.

7. Table 1 presents various health interventions evaluated in the program. To inform policy recommendations, it would be beneficial to discuss the individual contributions of each intervention. For instance, which intervention might have had a greater impact compared to others?

6. PLOS authors have the option to publish the peer review history of their article (what does this mean?). If published, this will include your full peer review and any attached files.

**Do you want your identity to be public for this peer review?** For information about this choice, including consent withdrawal, please see our Privacy Policy.

Reviewer #1: No

---

## [Decision Letter · Decision Letter 1]

18 Jan 2024

Evaluation of a novel approach to community health care delivery in Ifanadiana District, Madagascar

PGPH-D-22-01753R1

Dear Finnegan,

We are pleased to inform you that your manuscript 'Evaluation of a novel approach to community health care delivery in Ifanadiana District, Madagascar' has been provisionally accepted for publication in PLOS Global Public Health.

Best regards,

Abraham D. Flaxman, Ph.D.

Academic Editor

Reviewer Comments (if any, and for reference):

Reviewer's Responses to Questions

**Comments to the Author**

1. If the authors have adequately addressed your comments raised in a previous round of review and you feel that this manuscript is now acceptable for publication, you may indicate that here to bypass the “Comments to the Author” section, enter your conflict of interest statement in the “Confidential to Editor” section, and submit your "Accept" recommendation.

Reviewer #1: All comments have been addressed

2. Does this manuscript meet PLOS Global Public Health’s publication criteria? Is the manuscript technically sound, and do the data support the conclusions? The manuscript must describe methodologically and ethically rigorous research with conclusions that are appropriately drawn based on the data presented.

Reviewer #1: Yes

3. Has the statistical analysis been performed appropriately and rigorously?

Reviewer #1: Yes

4. Have the authors made all data underlying the findings in their manuscript fully available (please refer to the Data Availability Statement at the start of the manuscript PDF file)?

Reviewer #1: No

5. Is the manuscript presented in an intelligible fashion and written in standard English?

Reviewer #1: Yes

6. Review Comments to the Author

Reviewer #1: (No Response)

7. PLOS authors have the option to publish the peer review history of their article (what does this mean?). If published, this will include your full peer review and any attached files.

**Do you want your identity to be public for this peer review?** For information about this choice, including consent withdrawal, please see our Privacy Policy.

Reviewer #1: No
